

# Burden of COVID-19: a preliminary analysis in the population of Saudi Arabia

Syed Mohammed Basheeruddin Asdaq[1], Syed Imam Rabbani[2], Mohammed Kanan Alshammari[3], Reem Saud Alshammari[4], Mehnaz Kamal[5], Mohd Imran[6], Noufah Aqeel AlShammari[7], May Faiz Al Twallah[8] and Abdulmjeed Hussain Alshahrani[7]

[1] Pharmacy Practice, Almaarefa University, RIYADH, Riyadh, Saudi Arabia
[2] Department of Pharmacology and Toxicology, College of Pharmacy, Qassim University, Buraydah, Saudi Arabia
[3] Department of Pharmaceutical Care, Rafha Central Hospital, Rafha, Saudi Arabia
[4] Department of Pharmaceutical Care, Maternity and Children Hospital, Rafha, Saudi Arabia
[5] Department of Pharmaceutical Chemistry, College of Pharmacy, Prince Sattam Bin Abdulaziz University, Al-Kharj, Saudi Arabia
[6] Department of Pharmaceutical Chemistry, Faculty of Pharmacy, Northern Border University, Rafha, Saudi Arabia
[7] Security Forces Hospital, Riyadh, Riyadh, Saudi Arabia
[8] Northern Area Armed Forces Hospital, Hafar albaten, Saudi Arabia

Corresponding author
Syed Mohammed Basheeruddin Asdaq, sasdag@mcst.edu.sa

## ABSTRACT

**Background**. Coronavirus infection (COVID-19) has resulted in an unprecedented number of human deaths and economic losses. Analyzing the role of disease in different groups of people is useful for determining the burden of disease. As a result, the purpose of this study was to investigate the influence of COVID-19 on the Saudi Arabian population's quality of life, with a particular emphasis on the likely fall in their life expectancy.

**Methods**. A cross-sectional and retrospective analysis of 2,988 patients' databases was performed to assess COVID-19-induced mortality and complications in the community. The data was gathered from official websites that track the disease's impact daily between July and October 2021. On the acquired data, disability-adjusted life years (DALYs) and relative risk analysis were performed. The data was statistically analyzed using SPSS IBM 25. The Pearson's correlation test was used to examine the relationship between age and disease impact. The significance of the findings was determined by using a P value of less than 0.05.

**Results**. The data from the study indicated that the positive test rate, infection rate, and mortality rate in the population were 1.84% [+0.11/-0.39 of 95% confidence interval (CI)], 1.54% (+0.38/-0.52 of CI), and 1.59% (+0.4/-0.7 of CI), respectively. Highest percentage of mortality was observed in Riyadh (17%), followed by Jeddah (8.7%) and Makkah (7.5%). The DALYs/100,000 inhabitants increased progressively as the age of the population increased, and the highest value was found for those over 70 years old (25.73 ± 2.09). Similarly, the risk outcome (55%) increased significantly ($p = 0.037$) from 40 years onwards, and the maximum was observed at above 70 years (184%, $p = 0.006$). The correlation analysis indicated a significant association

($p = 0.032$) between age and COVID-19 induced mortality from the 40-year-old population onwards.

**Conclusion**. The current study found that the COVID-19 load in Saudi Arabia was comparable to that in nations that were said to have performed well during the pandemic. DALYs increased from 40 years to 60 years, although people over 60 years had a lower life expectancy and were more susceptible to infection. After 60 years, the occurrence of numerous co-morbid illnesses may have added to the population's burden of COVID-19. Further research in this area may yield a more precise estimate of the COVID-19-induced burden on the entire population.

# INTRODUCTION

Saudi Arabia is one of the seven countries that make up the Gulf Cooperative Council (GCC) group of nations. The country is the largest both in terms of geography and population and is located between the Persian Gulf and the Red Sea (*Alahmadi & Atkinson, 2019*). Human habitats are distributed throughout the country but mostly concentrated in cities and towns such as Riyadh, Damman, Makkah, Taif, Jeddeh, and Madinah. Although the population in these places includes mostly Saudi citizens, people from other nationalities who form the workforce can also be found in a significant number (*Asharaf & Alshekteria, 2008*).

In March 2020, the first case of coronavirus infection (COVID-19) was reported in Saudi Arabia's eastern region. Since then, the country has seen an outbreak of infection that has spread to every nook and corner (*Al-Tawfiq & Memish, 2020*). According to official data, the country was infected with COVID-19 in two waves. (https://www.moh.gov.sa/en/Ministry/MediaCenter/Publications/Pages/covid19.aspx). The first wave began in March 2020 and lasted until December 2020, while the second wave began in January 2021 and has been slow since August 2021 (https://covid19.moh.gov.sa/). The healthcare authorities took several proactive measures to prevent the spread of infection, and some of them included international travel bans, restrictions on social gatherings, compulsory wearing of facemasks, and maintaining social distance in public places. Entry to all public places, such as educational institutions, shopping malls, and hospitals, were restricted to those who failed to display government-approved health and vaccination status apps (*Al-Tawfiq & Memish, 2020*). A previous study suggests that the population of the country adopted these measures in a positive manner (*Al-Zalfawi et al., 2021*). These preventive measures were considered the most stringent among the region's nations, and they were largely linked to a decrease in COVID-19 positive cases and population mortality.

The COVID-19 pandemic has caused devastating effects not only on human health but also on the economy. Several people lost their livelihoods, in addition to their dear and near ones. According to a report, the unemployment rate during the pandemic peaked in

several parts of the world (*Deb et al., 2020*). Big business establishments closed during this period, causing irreparable damage to the gross domestic product (GDP). Previous studies have indicated that the rapid spread of COVID-19, its mortality, and complications have caused an enormous burden on mankind (*Bonaccorsi et al., 2020*).

The available data suggests that COVID-19 can cause infection in all age groups. In most patients, the infection is reported with mild symptoms such as fever, sore throat, and body aches. The severity and complications depend on several factors, and important among them are the age and co-morbid conditions of patients (*Zhou, 2020*). Estimating the mortality rate, case-fatality ratio, and crude-fatality rate are reported to provide a direct measure of the impact of a disease on the population. However, these methods do not determine the loss of life expectancy due to either mortality or disability (*Gaunt et al., 2011*). The burden of COVID-19 due to loss of life or complications has been reported to affect the life expectancy of the population. The global burden of a disease is an important measure for calculating the loss of quantity and quality of life. This can be calculated in several ways, the most important being disability-adjusted life years (DALYs). DALYs measure the number of years of life lost due to a disease or due to its complications. It is frequently used for both quantifying and comparing the burden of disease in a population (*Nurchis et al., 2020*).

A study conducted in sixteen European nations indicated a total loss of 4354 DALYs per 100,000 habitats. The data suggested that Italy, Sweden, Czechia, and the Netherlands suffered the maximum loss due to COVID-19, while the least effect was found in Estonia and Finland (*Al-Aly, Xie & Bowe, 2021*). The DALY data is also reported to indicate the implications of the prevention or treatment strategies adopted by healthcare providers for a particular disease (*Jo et al., 2020*). According to reports, Saudi Arabia is the second most infected country in the Gulf region with COVID-19 (*Alharbi et al., 2021*). Since analyzing the infection rate and mortality might not be sufficient to indicate the burden of COVID-19, this study was planned to evaluate DALY and relative risk of infection as parameters of the burden of disease in Saudi Arabia.

## MATERIALS AND METHODS

### Study data

The comprehensive COVID-19 data sheets published on the official websites of the ministry of health (MOH) (https://www.moh.gov.sa/en/Pages/default.aspx), the COVID-19 Dashboard (https://covid19.moh.gov.sa/), and Waqeya (https://covid19.cdc.gov.sa/) were retrieved. These sites are linked to the official WHO home page. Cumulative data on the total number of reverse transcriptase-polymerase chain reaction (RT-PCR) tests conducted, the number of COVID-19 positive cases detected, and mortality due to infection was recorded in an excel sheet from March 2020 to October 2021. The general authority of statistics, the ministry of health, and published articles gathered data on population size, population density, age, and the number of COVID-19 infections (*Al-Tawfiq & Memish, 2020*). The Ministry of Health's Health Electronic Surveillance Network (HESN) was also used to retrieve data from patients treated for COVID-19. This system collected data on

the demographic characteristics of COVID-19 positive patients from various parts of the country. The burden of disease was calculated using data from 2988 patients obtained from various sources between July 2021 and October 2021. This number is part of the total infection rate reported in the country. The following is the region-wise distribution of COVID-19 positive cases chosen for this study: Riyadh – 834, Makkah – 502, Jeddah – 439, Madinah – 306, Dammam – 217, Taif – 167, Buraidah – 107, Khobar – 103, Abha – 72, Dahran – 51, and the rest from other parts of the country.

## Descriptive analysis

The data collected from the previous mentioned sources was analysed in the following ways:

- *Positive test rate:* The total number of tests conducted in the country to determine the presence of COVID-19 in the population and the number of confirmed cases detected were utilized to find the positive test rate (*Rothan & Byrareddy, 2020*). Positive test rate = Positive tests/Total tests $\times 100$

- *Infection rate:* The total number of confirmed COVID-19 cases detected in the whole population of the country was represented as the infection rate (*Jo et al., 2020*).

- *The mortality rate:* The total number of deaths recorded in the whole population due to COVID-19 was calculated from the total number of confirmed COVID-19 cases (*Baud et al., 2020*).

- *Case-fatality ratio (CFR):* This value represents for the whole population affected with COVID-19 and was calculated using the formula described by *Onder, Rezza & Brusaferro (2020)*.

    CFR = (Total number of deaths X 100) / Total number of confirmed cases

- *DALY analysis:* The DALY was calculated using the method described by Murray and Lopez in 2020, and it is the sum of years of life lost (YLL) + years lost due to disability (YLD) (*Murray, 1994*). DALYs are used to measure the impact of a specific health condition on the population due to both premature death and morbidity. According to the guidelines, one DALY is considered as loss of 1 year of healthy life. All the complications of a disease state can be incorporated while calculating the DALY (*Gibney et al., 2013*). YLL was calculated from the number of COVID-19 related deaths reported for each age group in the population and the life expectancy lost at the time of death. The YLD was calculated from the total number of COVID-19 positive cases for each group of the population, multiplied by the loss of expected years due to disability and the disability weight. The value mentioned for the disability weight due to COVID-19 in the literature was used for YLD (*Gianino et al., 2021*). In our study, the recorded data of COVID-19 patients was divided into different groups depending on age (numbers) such as 0–9 (18), 10–19 (53), 20–29 (289), 30–39 (318), 40–49 (506), 50–59 (419), 60–69 (720) and above 70 years (665). After analysis, the DALY data was represented per 100,000 inhabitants according to the population size of the country as indicated by the general authority of statistics.
- *Relative risk (RR) analysis:* This compares the likelihood of mortality for a specific group of people to the risk of death for the entire population (*Mathers et al., 2006*). It is calculated from the formula.

RR = Risk in one group/Risk in all the other groups.

Further, risk outcome was calculated based on the value of RR and as per the procedure described by *Pijls et al. (2021)*.

## Representation of data

The data recorded for the positive test rate, infection rate, and mortality rate was represented as percentages graphically (Fig. 1). Further, the total mortality recorded in five major cities in Saudi Arabia is indicated as a total number (Fig. 2). The data from the DALY analysis was represented in tabular form (Table 1) as YLL, YLD, DALYs, and DALYs per 100,000 inhabitants. The analysis of relative risk was represented as CFR, RR, and risk outcome in different groups of the population in tabular form (Table 2).

## Ethical clearance

The data was conducted after obtaining ethical clearance from the research committee of the College of Pharmacy (MCST/COP #20/2021), AlMaarefa University, Riyadh, Saudi Arabia.

## Statistical analysis

The data obtained from the study was statistically analyzed by Statistical Package for the Social Sciences (SPSS, IBM 25) software. The data collected from 2988 patients was categorized into different age groups. The sample size in different age-groups varied between 69 and 441. Due to the requirements of the statistical software, incomplete data from approximately 107 patients was not considered. In addition, this number was excluded from the 2,988 patient records that were used for the analysis. The data for each group of patients was entered and analyzed for variables such as years of life lost (YLL), years lost due to disability (YLD), case fatality ratio (CFR), relative risk, and the outcome of risk. Both parametric and non-parametric tests were utilized for the statistical analysis. The one-way analysis of variance (ANOVA) was used to test the hypothesis that population distribution influences disease outcome. In this objective, the influence of the age of the population was tested over the outcome of COVID-19 and a comparison was drawn between groups. A non-parametric test such as Mann–Whitney was used to analyze the second objective, that the outcome of the COVID-19 is independent of the population's distribution. Further, the Pearson coefficient of correlation was used to determine the association between two variables, such as age and mortality due to COVID-19. While analyzing the data, a fixed confidence interval (lower–95% and upper–95%) was used to determine the uncertainty using the values obtained from sample mean, standard deviation, and sample size (*Singh, Devleesschauwer & Khatkar, 2022*). A $P$-value of less than 0.05 was considered to determine the significance of all the analysis.

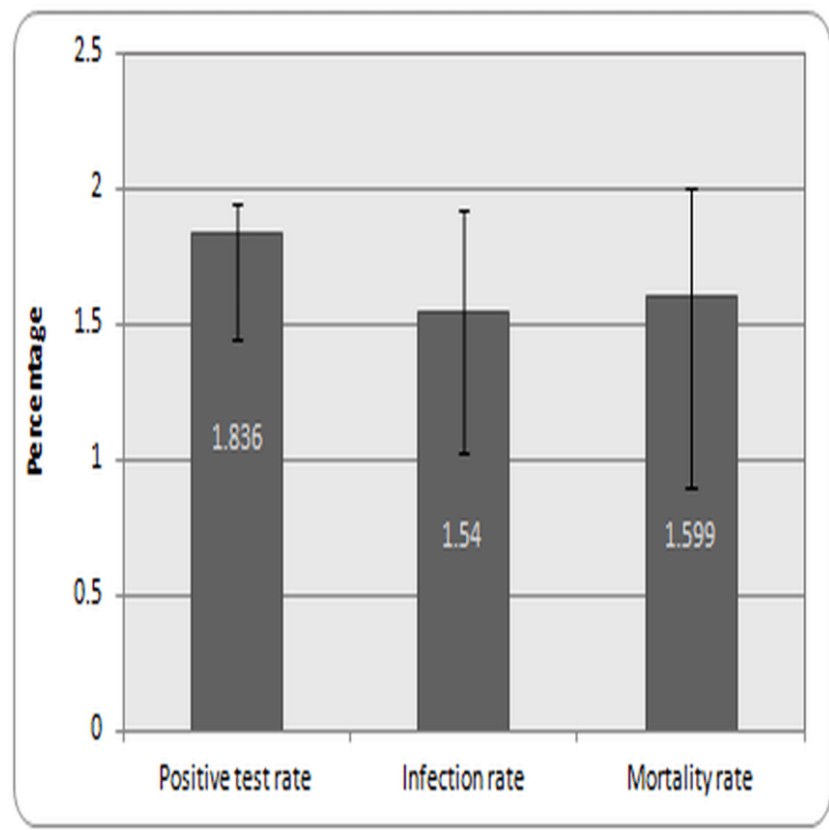

**Figure 1  Infection and mortality rate in Saudi Arabia.** Note: Values are expressed as rate in population with +/- 95% CI.

## RESULTS

### Infection rate and mortality rate due to COVID-19 in Saudi Arabia

Figure 1 represents the values of the positive test rate, infection rate, and mortality rate. A total of 29,841,399 COVID-19 tests were conducted on the population on October 18, 2021. The total number of COVID-19 positive cases detected was 5,47,969, and therefore the positive test rate was found to be 1.836%. Saudi Arabia has a population of 35,498,886 people and the infection rate in them was found to be 1.54%. The total mortality recorded in the COVID-19 diagnosed patients was found to be 8,765 and hence the mortality rate was calculated as 1.599%.

### Disability adjusted life years (DALYs) due to COVID-19 in Saudi Arabia

The DALYs calculated by the addition of YLL and YLD indicated the lowest for the 0–9-year age group. The DALYs per 100,000 habitants aged 0–9 years old were found to be 0.77. This value progressively increased to 2.32 in 10–19 years, 4.05 in 20–29 years, and 7.63 in 30–39 years. From 40–49 years of age, DALYs increased to double digits and were found to

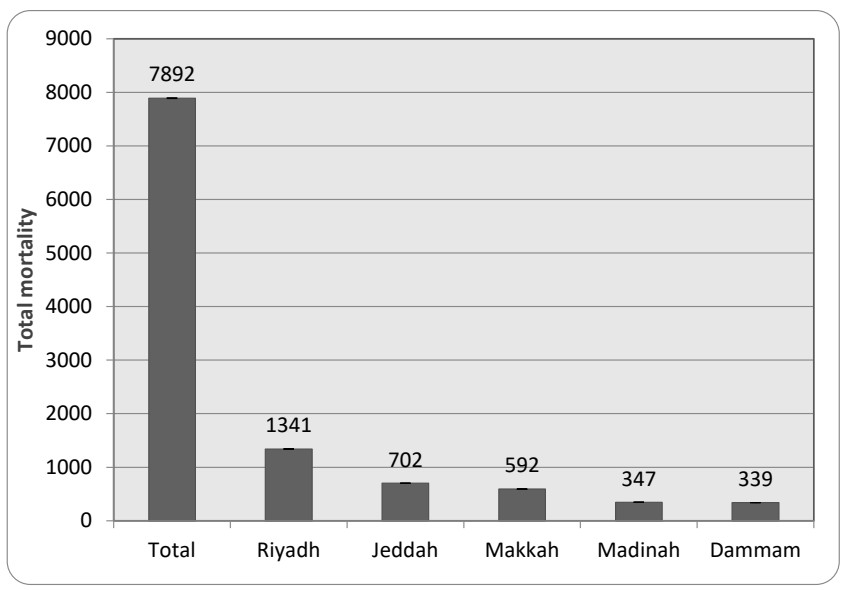

**Figure 2** **Total mortality in major cities of Saudi Arabia.** Note: Values represent the total mortality with +/- 95% CI.

**Table 1** Disability adjusted life years (DALYs) due to COVID-19 in Saudi Arabia.

| Age group | YLL | YLD | DALYs | DALYs /100,000 |
|---|---|---|---|---|
| 0–9 Yrs | 181 ± 4.33 | 95 ± 5.21 | 276 ± 11.47 | 0.77 ± 0.02 |
| 10–19 Yrs | 696 ± 21.84 | 128 ± 7.64 | 824 ± 29.47 | 2.32 ± 0.07 |
| 20–29 Yrs | 876 ± 26.52 | 565 ± 22.36 | 1441 ± 38.25 | 4.05 ± 0.12 |
| 30–39 Yrs | 1682 ± 37.86 | 1028 ± 35.94 | 2710 ± 54.26 | 7.63 ± 0.36 |
| 40–49 Yrs | 2167 ± 43.28 | 1768 ± 38.67 | 3935 ± 59.62 | 11.08 ± 0.62 |
| 50–59 Yrs | 2904 ± 44.39 | 2373 ± 42.58 | 5277 ± 71.08 | 14.86 ± 0.81 |
| 60–69 Yrs | 4194 ± 60.21 | 3269 ± 50.98 | 7463 ± 84.96 | 21.02 ± 1.67 |
| Above 70 Yrs | 5018 ± 64.26 | 4119 ± 57.46 | 9137 ± 91.68 | 25.73 ± 2.09 |

**Notes.**
Values are expressed as Mean ± SD.
YLL, Years of life lost; YLD, Years lost due to disability.

be 11.08. For 50–59 years, it was found to be 14.86 and for 60–65 years, it was 21.02. The highest recorded DALY was observed for people over 70 years old (25.73) (Table 1).

## Relative risk of CFR due to COVID-19 in Saudi Arabia

The relative risk observed in different groups of people is summarized in Table 2. The lowest CFR and negative risk outcome were found in the 0–9-year age group. As the age of the groups increased, the CFR as well as the RR and risk outcome also increased. A significant level of RR value (1.55, $P = 0.037$) was found for 40–49 years onwards. People of the age group 50–59 years were found to be at a 78% risk (RR = 1.78, $P = 0.043$), 60–69 years old at a 132% risk (RR = 2.32, $P = 0.008$), and the highest risk (184%) was observed in the over 70-year-old population (RR = 2.84, $P = 0.006$).

**Table 2  Relative risk (RR) of CFR due to COVID-19 in Saudi Arabia.**

| Age group | CFR | Lower 95% CI | Upper 95% CI | RR | Risk outcome | *p*-value |
|---|---|---|---|---|---|---|
| 0–9 Yrs | 0.002 | −0.001 | 0.005 | 0.05 | −95% | 0.351 |
| 10–19 Yrs | 0.011 | 0.008 | 0.046 | 0.12 | −88% | 0.690 |
| 20–29 Yrs | 0.126 | 0.098 | 0.138 | 0.61 | −39% | 0.298 |
| 30–39 Yrs | 0.291 | 0.174 | 0.311 | 1.02 | 2% | 0.086 |
| 40–49 Yrs | 0.928 | 0.416 | 1.192 | 1.55[*] | 55% | 0.037 |
| 50–59 Yrs | 1.269 | 0.869 | 1.485 | 1.78[*] | 78% | 0.043 |
| 60–69 Yrs | 2.112 | 1.702 | 2.988 | 2.32[**] | 132% | 0.008 |
| Above 70 Yrs | 3.025 | 2.677 | 3.966 | 2.84[**] | 184% | 0.006 |

**Notes.**

Relative risk analysis from COVID-19.

[*]$p < 0.05$.

[**]$p < 0.01$ compared between groups.

## Total mortality recorded in major cities in Saudi Arabia

Figure 2 indicates the five highest mortality recorded in the cities of Saudi Arabia. Riyadh, being the capital city and with the largest population, recorded the highest (17% of total mortality), followed by Jeddah (8.9%), Makkah (7.5%), Madinah (4.4%), and Dammam (4.3%). The first two cities are considered the largest in the kingdom in terms of human population and size.

## Summary of correlation between age of population and COVID-19 induced mortality

The association between the age of the population and the mortality due to COVID-19 indicated the lowest correlation coefficient for 0–9 years. The correlation increased to a positive level from 10–19 years and became significant from 40–49 years onwards. The correlation coefficient value for 40–49 year olds was 0.56 ($P = 0.032$), and it increased to 0.64 ($P = 0.044$) in 50–59 year olds. The highest correlation value was found in people aged 60–69 ($r = 0.72$, $P = 0.006$) and over 70 ($r = 0.78$, $P = 0.007$) (Table 3).

## DISCUSSION

The present study analyzed the burden of COVID-19 in Saudi Arabia. The region recorded a total of 5,47,969 COVID-19 cases till October 18th, 2021. The first case was detected in a citizen who came to the country from Persia. Following that, many positive cases were discovered in various parts of the country, and the infection rate is now considered the second highest after the United Arab Emirates (*Al-Tawfiq & Memish, 2020*).

The data from the study indicated that the infection rate in the whole population of the country is 1.54%, although the health authorities conducted more than 29,841,399 RT-PCR tests (positive test rate is 1.836%) on the population (Fig. 1). The values suggested that the preventive measures imposed by the healthcare and other government authorities restricted the spread of infection in the population (*Alkhowailed et al., 2020*). These values indicated similarities with other countries that were reported to have managed the infection effectively, such as South Korea, Germany, and New Zealand (*Naumann et al., 2020*).

**Table 3  Summary of Pearson correlation analysis between age and mortality due to COVID-19.**

| Age group | Correlation coefficient (r) | Lower 95% CI | Upper 95% CI | p-value |
|---|---|---|---|---|
| 0–9 Yrs | −0.24 | 0.152 | 0.179 | 0.622 |
| 10–19 Yrs | 0.16 | 0.246 | 0.392 | 0.981 |
| 20–29 Yrs | 0.31 | −0.236 | 0.096 | 0.265 |
| 30–39 Yrs | 0.38 | 0.095 | 0.108 | 0.723 |
| 40–49 Yrs | 0.56[*] | 0.221 | 0.462 | 0.032 |
| 50–59 Yrs | 0.64[*] | 0.146 | 0.197 | 0.044 |
| 60–69 Yrs | 0.72[**] | −0.016 | 0.042 | 0.006 |
| Above 70 Yrs | 0.78[**] | 0.104 | 0.211 | 0.007 |

**Notes.**

Pearson correlation analysis.

[*]$p < 0.05$.

[**]$p < 0.01$ compared between groups.

The World Health Organization (*WHO, 2021*) has recommended several measures to prevent the spread of the COVID-19 infection (https://www.who.int/emergencies/diseases/novel-coronavirus-2019). According to an earlier study, the attitude of the public plays an important role in achieving the objectives of the healthcare authorities (*Al-Zalfawi et al., 2021*). The data suggest that the people of Saudi Arabia responded in a positive manner to the preventive measures adopted by several authorities during the COVID-19 outbreak.

According to the literature, several factors, such as severity of infection, rate of infection, co-morbidities, socio-economic status, and healthcare facilities, could play an important role in the outcome of a diseased condition. These factors are reported to be affected at different chronological stages of a pandemic. One of the established methods to determine the burden of a disease in the affected population is to calculate the loss of life expectancy and can be done using DALY (*Nurchis et al., 2020*). Hence, DALY analysis was done on patients infected with COVID-19 in different provinces of Saudi Arabia. The DALY analysis of 2988 COVID-19 positive patients suggested that YLL (years of life lost) and YLD (years lost due to disability) due to viral infection increased with the age of the population (Table 1). Earlier studies suggested that YLL and YLD values provide important information not only about the disease but also the outcome of the treatment strategies against a diseased state (*Nurchis et al., 2020*). The progressive increase in DALYs values across age groups suggested a link between age and COVID-19-related complications (Table 1). A threefold increase in DALY in patients aged 60–65 years supports previous findings that COVID-19-induced mortality and complications are common in this age group (*Naumann et al., 2020*). The data from the present study also suggested that the total mortality was high in regions with dense populations (Fig. 2). As reported earlier, urbanization and advancing age carry the burden of several co-morbidities (*Al-Zalfawi et al., 2021*). The mortality rate and CFR indicated in Figs. 1 and 2, respectively is observed to be influenced by several factors. The average time of death due to COVID-19 is found to be independent of positivity rate reported in the population (*Xie et al., 2020*). The CFR values in the present study for patients above 60 years were found to be greater than 2.

These values are accordance with the previous results reported for people from Asian origin (*Cao, Hiyoshi & Montgomery, 2020*). People over 60 years of age are most likely to suffer from different cardiovascular, central nervous system, respiratory, renal, and carcinogenic diseases (*Nurchis et al., 2020*). The present data also revealed that there was no major increase in DALYs for people aged 70 and above in comparison with the 60–65-year group. The life expectancy in Saudi Arabia is reported to be 75.13 years (*Jo et al., 2020*) and the data from this study suggests that there was no major loss in terms of YLL and YLD in this population group (Table 1).

Further, the cumulative DALYs/100,000 was found to be 87.50. The relative risk and risk outcome are indicative of level and probability of disease occurrences in a group of population, respectively (*Liu et al., 2021*). The relative risk analysis due to COVID-19 indicated that case-fatality ratio and risk outcome increased according to the age of the population. A significant increase in RR values (RR = 1.55, $P = 0.037$) was observed from 40–49 years of aged groups along with rise in CFR (0.92). The highest CFR and RR values were found for the people above 70 years old (CFR = 3.02, RR = 2.84, $P = 0.006$) (Table 2). These values indicated similarity with previous study, where population with age 0–34 years had lower risk, but the risk increased after 35 years age (*Ritchie et al., 2020*). The DALY values when compared to the previous studies suggested that the health care facilities offered to the COVID-19 patients have influenced positively in minimizing both the mortality and complications. The data when compared falls between 59 (Finland) and 116 (Denmark), reported from those countries that have controlled effectively the COVID-19 pandemic (*Naumann et al., 2020*).

The mortality rate due to COVID-19 was found to be 1.566% (Fig. 1). This rate was found to be within the range (1–2%), similar to those seen in countries with lower mortality rates, but lower than the global average fatality rate ($\approx$3). According to WHO, the infection due to COVID-19 cause mild symptoms in most people, however, complications and deaths were common in patients who are aged and sufferers of chronic diseases. Accumulated data from the MOH resources suggests that 46% of Saudi Arabia's population suffers from cardiovascular disease. Incidences of these diseases were found to be more common in urban areas (*Hussein & Ismail, 2017*) and so the mortality was found to be high in major cities of Saudi Arabia (Fig. 1). There could also be other factors contributing to the time between infection and mortality from COVID-19. In one such study conducted in Spain, a delay up to 30 days was observed between the incubation period, time from onset of illness to death, and the case fatality ratio (*García-García, Vigo & Fonfría, 2021*). For effective interpretation of COVID-19 data from diverse places, a proper procedure such as determining the precise period between the onset of symptoms and mortality is required.

Obesity, diabetes, and hypertension was found to be frequent disorder amongst the patients diagnosed with metabolic diseases including the young population (*Caussy et al., 2020*). Obesity in the previous studies has been considered an important risk factor for severity due to COVID-19 (*Sattar, McInnes & McMurray, 2020*). A meta-analysis of COVID-19 patients indicated a strong relation between obesity and ICU admission after infection with COVID-19 (*Yang, Hu & Zhu, 2021*). These studies indicated that obesity increases the chances of other diseases such as diabetes mellitus, renal insufficiency, and

cancer, thus complicating the COVID-19 infection (*Caussy et al., 2020*; *Sattar, McInnes & McMurray, 2020*; *Yang, Hu & Zhu, 2021*).

In hypertensive patients, the rennin-angiotensin system is reported to be in a hyperactive state and the severe acute respiratory syndrome-coronavirus-2 (SARS-CoV-2) has a special affinity for angiotensin converting enzyme-2 (ACE-2). The virus uses them for entering the host and during COVID-19, the expression of ACE-2 was also found to be enhanced (*Zhang et al., 2020*). The likelihood of other co-morbidities in hypertensive patients reported to aggravates the complications of COVID-19 (*Fang, Karakiulakis & Roth, 2020*). Apart from these, the prevalence of cancer (10%), diabetes mellitus (5%), and respiratory diseases (3%) were also reported in Saudi population (*Hussein & Ismail, 2017*). The implication of these diseases has been reported in the COVID-19 induced complications by several mechanism including altered immunological reactions (*Fang, Karakiulakis & Roth, 2020*).

The prevalence of these complications has been reported to rise with population age (*Hussein & Ismail, 2017*). This correlation between age and an increased mortality can be observed in Table 3. The data from this analysis indicated that the correlation coefficient between age and mortality due to COVID-19 increased significantly ($P < 0.05$) from 40 years onwards and a strong correlation ($r = 0.72$) can be observed in a population of over 60 years of age. The available data suggested that healthcare facilities in Saudi Arabia follow the standard updated guidelines of WHO while treating the COVID-19 patients. All the regions of the country have designated hospital to manage the complications of COVID-19. Besides, the COVID-19 vaccination programme was reported to have covered more than 70% of population, achieving the herd immunity (https://covid19.moh.gov.sa). The proactive, preventive, and established therapeutic interventions implemented by healthcare providers could be linked to the management of current pandemic situation as well lower CFR among the population of Saudi Arabia. More research in this direction involving a greater number of patients might provide clarity on the actual burden of COVID-19 in the country.

## LIMITATION OF THE STUDY

Although we used a representative sample size to examine the COVID-19 burden in the Saudi Arabian population, we advise caution in generalizing this information without further validation in a larger sample size. The wider confidence intervals (95%) observed in this study can be resolved if more studies are conducted using the tested parameters. Moreover, the influence of the pandemic in causing the burden on the other demographic characteristics of the population, such as race, gender, education, and nationality, also needs evaluation including the age-adjusted infection and positivity rate.

## CONCLUSION

Analysis of 2988 COVID-19 patients' disability adjusted lost years (DALYs) revealed that the values increased in those over 40 years. Patients over the age of 60 had a greater loss of life expectancy, and relative risk measurement demonstrated that this group of patients is at a higher risk of mortality due to COVID-19. Considering the preliminary data, a

new strategic plan is required from the healthcare providers to reduce the burden on the population. In the absence of effective therapeutic intervention and emergence of several variants of SARS-CoV-2, strategies must be focused on reducing the prevalence of co-morbidities in the population. More studies might determine the actual burden on the population with different demographic characteristics in the event of a prolonged COVID-19 pandemic.

### Funding

Syed Mohammed Basheeruddin Asdaq received financial support (TUMA-2021-1) from AlMaarefa University, Riyadh, Saudi Arabia to do this research. The funders had no role in study design, data collection and analysis, decision to publish, or preparation of the manuscript.

### Grant Disclosures

The following grant information was disclosed by the authors:
AlMaarefa University, Riyadh, Saudi Arabia.

### Competing Interests

The authors declare there are no competing interests.

### Author Contributions

- Syed Mohammed Basheeruddin Asdaq conceived and designed the experiments, performed the experiments, prepared figures and/or tables, and approved the final draft.
- Syed Imam Rabbani conceived and designed the experiments, performed the experiments, analyzed the data, prepared figures and/or tables, and approved the final draft.
- Mohammed Kanan Alshammari conceived and designed the experiments, performed the experiments, analyzed the data, authored or reviewed drafts of the paper, and approved the final draft.
- Reem Saud Alshammari conceived and designed the experiments, analyzed the data, authored or reviewed drafts of the paper, and approved the final draft.
- Mehnaz Kamal conceived and designed the experiments, analyzed the data, prepared figures and/or tables, and approved the final draft.
- Mohd Imran conceived and designed the experiments, performed the experiments, analyzed the data, prepared figures and/or tables, and approved the final draft.
- Noufah Aqeel AlShammari conceived and designed the experiments, performed the experiments, authored or reviewed drafts of the paper, and approved the final draft.
- May Faiz Al Twallah conceived and designed the experiments, performed the experiments, prepared figures and/or tables, and approved the final draft.
- Abdulmjeed Hussain Alshahrani conceived and designed the experiments, analyzed the data, prepared figures and/or tables, and approved the final draft.

## Ethics

The following information was supplied relating to ethical approvals (i.e., approving body and any reference numbers):

Research committee of College of Pharmacy (MCST/COP#20/2021), AlMaarefa University, Riyadh, Saudi Arabia.

## Data Availability

The raw data is available at:

- Kingdom of Saudi Arabia's Ministry of Health: https://www.moh.gov.sa/en/Pages/default.aspx

- Ministry of Health Dashboard: https://covid19.moh.gov.sa/

- Public Health Authority: https://covid19.cdc.gov.sa/.

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

## FURTHER READING

**Ministry of Health of Saudi Arabia. 2021a.** COVID-19 infection rate in Saudi Arabia. *Available at* https://www.moh.gov.sa/en/Ministry/MediaCenter/Publications/Pages/covid19.aspx (accessed on 15 October 2021).

**Ministry of Health of Saudi Arabia. 2021b.** COVID-19 vaccination update in Saudi Arabia. *Available at* https://covid19.moh.gov.sa (accessed on 18 October 2021).