# Peer review of "Burden of COVID-19: a preliminary analysis in the population of Saudi Arabia"

_PeerJ, doi:10.7717/peerj.13219_

## Round 0.1 · original submission · Major Revisions

Both reveiwers kindly provided constructive comments for improvement. In particular, one of reveiwers noted that the title of this paper is on the "burden" of COVID-19, but the authors mainly handled "case count" rather than mortality or hospitalizations. Also, the absence of uncertainty (e.g. 95% CI) has been noted.

Reviewer 1 ·

Basic reporting

To estimate the disease burden, the DALY is perhaps one of the most reasonable indicators. Therefore, I agree with the focus point of the authors in the manuscript. However, there are many points to be improved and explained. First of all, all abbreviations must be fully described before they are used (e.g., MOH). The number of digits should be consistent in the manuscript, especially in the result section. Some bold and italic expressions are unformal; therefore, I would like the authors to send the manuscript to an English editing service. The main aim of the manuscript is to estimate DALYs by age groups in Saudi Arabia; however, I do not understand the relationship between other measures like infection rate and DALYs, and those points are not explained in the discussion well. Also, although positive rate, infection rate and mortality are analyzed in the study, they are quite influenced by the time of epidemic but the authors tried to discuss using only point estimates of them without any uncertainties. Overall, the manuscript may not meet the standards of the journal.

Experimental design

The DALYs are calculated using 2988 samples from all confirmed COVID-19 cases if I correctly read the manuscript. However, I am confused about calculating the positive rate, infection rate, mortality, and CFR. Are they calculated from 2988 samples or all cases? In addition, were those indicators adjusted by age?
I would like the authors to explain the sampling process more clearly; otherwise, the readers may lose the critical points.
I do not follow the explanation of 2.2 descriptive analysis. For example, is the positive rate the total number? Please correctly describe each measure, also use the formula if you want.
If the authors estimate CFR using the latest datasets, the time delay between infection (confirmation) and death is not considered. In this case, CFR could be underestimated. Would you please describe this point somewhere or tackle this?
The relative risk for what? (many sentences are very vague)?
In the manuscript, mortality and CFR were mixed and misused. For example, table 2 shows RR of mortality, but the table shows CFR. Are they the same?
Figure 1 and 2 could be improved if the authors add the uncertainties (confidence intervals). In figure 2, the difference in mortality between cities is more important than proportion.
The details of the age groups are not described in the manuscript. Would the authors please explain that in the method?
I think 2.3 should be in the result.
L192. What are two variables?

Validity of the findings

In table 2, CFRs appeared to be smaller, especially older people, compared with other countries. How reliable are estimated values? Is that because of the underreported deaths due to COVID-19? In addition, confidence intervals do not seem to match with CFRs.
I do not understand what risk outcome stands for.
What do the authors think the impact of COVID-19 vaccines on the result?
Considering the many indicators (e.g., mortality), they must be influenced by the epidemic and vaccination progress in the population. Therefore, I would like to suggest that this point be tackled in the discussion.
What is the significance level of two-tailed t test in table3? Are they different from p-values?
As mentioned earlier, transmission dynamics are changed according to each stage of the epidemic. Therefore, it may be precarious to estimate DALYs using the entire epidemic without uncertainty. Rather than that, it would be good to show them different epidemic stages.

Additional comments

I would recommend that the details of the study period be explained in the abstract.

·

Basic reporting

This paper reports the results of a descriptive study of several measures of the burden associated with the COVID-19 pandemic in Saudi Arabia.

On the whole, the article is written lucidly and the quality of written English is fair. Literature references have been provided for most of the assertions made in the text; however, the quality of the provided figure (Figure 1) is suboptimal in terms of resolution, background and captioning and does not add much to the text (as it simply repeats the raw values of three COVID-19 indices which are not in any way directly comparable).

The introduction would benefit from a greater depth of coverage of the various measures used to assess the burden of COVID-19 at a regional and national level and their strengths and limitations. The authors could also provide some background information on the government response to COVID-19, public adherence to the same, and how effective this response may have been in curtailing the spread of the disease.

The authors could also consider using a more visually informative illustration, such as a coloured or shaded map, to indicate the number / prevalence / fatality rates and DALY for COVID-19 in different regions of the country.

Experimental design

The authors have used a variety of indices to capture various aspects of the burden of COVID-19. In their estimation of disability-adjusted life years (DALY), they claim to have used a value derived from earlier literature and have cited a reference for this (no. 9); however, the concerned paper does not contain any guideline value with which DALY could be calculated. A correct citation along with more explicit details of the calculation should be provided, as this is one of the novelties of the current research.

The authors have mentioned collecting COVID-19 data from official governmental web sites. What was the degree of agreement between the different sources cited by them? Were the figures obtained consistent with those available on international data aggregators (e.g. the Johns Hopkins Medical University resource centre / dashboard for COVID-19?) This could be valuable in terms of ensuring the validity of this data.

The authors have examined the relationship of age with mortality. Though their method of data collection necessarily precludes the collection of more specific individual-level information (such as comorbid medical conditions or obesity) it would be useful if the authors could assess the relationship between gender and mortality, which has been documented in several other countries and regions. (If this data is not available, this should be acknowledged as a limitation).

Validity of the findings

As mentioned above, the authors should provide clearer details of the calculation of DALY, as this is the only novel measure in this current study and the citation provided appears to be incorrect.

The results are somewhat simplistic in nature (being largely descriptive) and no inferential analysis was carried out other than a possible correlation between age and mortality.

Further, it is not clear how the correlation between age and mortality was carried out. The authors have referred to Pearson's test in section 2.5, but have presented their results in ordinal age blocks (from 0-9 to >70) for which Pearson's test would not be appropriate (a Spearman rank correlation or other nonparametric method would be advisable in this case).

The discussion of the renin-angiotensin system and of medical comorbidities in Section 4 is somewhat tangential to the current paper (where these parameters have not been assessed) and can be shortened or deleted.

The authors have made the assertion that their results are comparable to those of "well-managed" countries. Is there any objective standard (for example, a statistical comparison of COVID indices, or guidelines provided by the WHO) of what would constitute "well-managed"? If not, this statement can be replaced with a more balanced assertion, such as "this rate was similar to those seen in countries with lower mortality rates" or "this rate was n standard deviations below the global average mortality rate".

The conclusion merely restates the key findings of the study. Instead, it should highlight the public health significance of the findings reports and their future implication, and should also acknowledge that DALYs may need to be recalculated in future given the phenomenon of "long COVID" being reported from other countries.

Additional comments

None.

---

## Round 0.2 · Minor Revisions

Please very carefully address comments from the reviewer 1. Uncertainties (i.e. 95% confidence intervals) are requested, and other technical comments have been provided.

Reviewer 1 ·

Basic reporting

Thank you for tackling my comments. However, there are still many concerns in the manuscript.

Experimental design

no comment

Validity of the findings

- Please consider adding the uncertainty of each outcome in table 1.
- Please add the uncertainties in the abstract, for instance, positive rate.
- What do the values inside the parentheses concerning mortality represent in the abstract?
- Although I pointed out the concern about the delay between infection and death in the previous review, the authors did not mention it. Therefore, please discuss the impact of the estimation of CFR in real-time. Estimated CFR may be underestimated due to the delay in reporting deaths.
- The authors cited the paper written by Jo et al. (2020) to calculate the confidence interval. Unfortunately, I did not find a detailed calculation of the confidence interval in that paper. Please describe how the authors estimated the 95% CIs.
- Please add legends in all figures (e.g., explanation of error bars).
- Please consider showing the number of each age group in the method part or table.
- In the manuscript, I am confused about “mortality” and “mortality rate”. Please clarify the difference between both of them if they are used peculiarly.
- To my understanding, mortality is commonly represented as rate. I am afraid that figure 2 may be incorrect because Y-axis seems strange.
- Please check 95% CIs in table2 because they are not matched with estimated CFR (maybe %?).
- The authors discuss the estimated CFR by age group using the paper written by Cao Y et al. from L296. However, it does not make sense because CFR among a particular age group is compared with that adjusted by age, 3.31%. In this case, the authors should estimate CFR in Saudi Arabia adjusted by age because CFR is highly influenced by age. Also, the authors mentioned the impact of vaccination on the population, so they need to consider the difference of periods. The reason that I was concerned about the reliability of the data last review was CFR among older people appeared to be lower than that in other countries. Considering the vaccination in Saudi Arabia, the result may be reasonable. However, I am afraid I have to disagree that the authors argue that the estimated CFRs are reasonable when comparing CFR among certain age groups with CFR adjusted by age.
- Please estimate the age-adjusted positive rate, infection rate and mortality rate. If the authors could not, please mention this as one of the limitations.

Additional comments

no comment

·

Basic reporting

The authors have incorporated the suggestions made in the earlier review regarding providing a clearer background for this study.

Experimental design

Clarifications regarding methodology have been provided. These were verified and found acceptable.

Validity of the findings

In view of the clarifications provided by the authors above (see #2), I have no concerns about the findings in this area.

Additional comments

None.

---

## Round 0.3 · accepted · Accept

The reviewer confirmed that you have successfully addressed earlier comments.

Reviewer 1 ·

Basic reporting

Thank you for addressing my comments on the manuscript; I think it was well revised and easy to follow the contents.

Experimental design

no comment

Validity of the findings

no comment

Additional comments

no comment